# Transgenerational effects of grandparental and parental diets combine with early-life learning to shape adaptive foraging phenotypes in *Amblyseius swirskii*

Peter Schausberger [1✉] & Dalila Rendon[1]

Transgenerational effects abound in animals. While a great deal of research has been dedicated to the effects of maternal stressors such as diet deficiency, social deprivation or predation risk on offspring phenotypes, we have a poor understanding of the adaptive value of transgenerational effects spanning across multiple generations under benign conditions and the relative weight of multigenerational effects. Here we show that grandparental and parental diet experiences combine with personal early-life learning to form adaptive foraging phenotypes in adult plant-inhabiting predatory mites *Amblyseius swirskii*. Our findings provide insights into transgenerational plasticity caused by persistent versus varying conditions in multiple ancestral generations and show that transgenerational effects may be adaptive in non-matching ancestor and offspring environments.

[1] Department of Behavioral and Cognitive Biology, University of Vienna, Djerassiplatz 1, 1030 Vienna, Austria. ✉email: peter.schausberger@univie.ac.at

Transgenerational effects are non-genetic environmental influences that are passed from ancestral generations to offspring and typically cause phenotypic variation but no variation in the DNA sequence[1,2]. Such effects have substantial ecological and evolutionary implications[1,3–7] and are assumed to be selectively favored if environments vary spatially and/or temporally across generations, reliable cues allow ancestors to anticipate offspring environments, and phenotypic plasticity has low costs[3,4,6,8,9]. Transgenerational effects represent communication of information from ancestors (senders) to offspring (receivers)[7] and may be mediated by a diversity of, mutually non-exclusive and often interrelated, proximate mechanisms including epigenetic, physiological, morphological, neurological, and/or behavioral changes[5,10]. They may span across an indefinite number of generations yet the most widely studied types are those transmitted by mothers and/or fathers to offspring (maternal and paternal, or more generally, parental effects). While parental, and here especially maternal, effects are quite well researched and documented in numerous contexts within plant and animal taxa (for review[3–5,8–10]), we have little understanding of how effects across multiple generations, such as the interaction of grandparental and parental experiences, combine with personal experiences to produce adaptive transgenerational outcomes in animals[7,11–14]. The adaptive value of transgenerational effects for offspring strongly depends on their persistence. For some effects, and under some circumstances, it might be favorable to persist or be transferred over several generations or to be updated and adjusted with each generational switch. While it is widely assumed that many of these effects are transient and rarely transmit over several generations, pertinent studies are documenting morphological, physiological, behavioral, and/or biochemical changes persisting over multiple generations[7,9,15–21]. Knowing about the prevalence and persistence, reversibility, and modifiability of transgenerational effects are critical for determining their adaptive value[9].

Numerous studies have focused on the effects of parental stressors such as nutritional deficits, toxins or malnutrition (mice[22], humans[23], lizards[24], birds[25,26], fruit flies[27]), social conditions (fish[28], bumblebees[29]), and disease/predation risk (fish[30–32], snails[33], fruit flies[34]) on offspring life history, physiology, and behavior including learning. Compared to stressors and experience of traumatic events, insights into transgenerational responses by offspring to benign conditions or conditions with little stress in ancestral generations are limited, except for transgenerational transmission of diet preference or bias (for review[9]). Clearly, more in-depth studies are needed to promote our understanding of adaptive transgenerational plasticity under benign conditions, given that this is the usual state. Effects of maternal diet or rearing conditions on diet preference or bias of offspring have been documented for many animal taxa (often under the heading prenatal or embryonic learning). For example, mammals such as rabbit young[35], piglets[36], dogs[37], and humans[38] may be cued to maternal diets. Odor learning and ensuing odor preference by larvae of the parental generation is transmitted to offspring in the butterfly *Bicyclus anynana*[39]. Predatory mite mothers, *Neoseiulus californicus*, may transmit prey odor experiences to their offspring, which then display a preference for these odors[40]. However, studies focusing on multigenerational effects of grandparental and parental diet experiences (other than nutritional deficits or malnutrition; see for example Deas et al. 2019[13]) and intergenerational diet switches vs. consistency on foraging performance and learning by offspring, and associated fitness implications, are lacking[41]. In fluctuating environments, information from two consecutive generations passed on to offspring may be more substantial, because having been collected over a longer time span, rather than only parental information.

Such information may even be additive or synergistic as opposed to a partial or complete update or removal of previous information with every new generation. Also, multigenerational information increases the chance of a match between ancestor and offspring environments[9]. Though it is expected that grandparental and parental information have different weights for offspring, the weights depend on which environment the offspring experience. With information from multiple ancestral generations, offspring may be better prepared to cope with one of several possible situations or the combination thereof. Offspring carry the information from their parents, modulate/update their information state by personal experience and pass this modulated/updated information on to their offspring.

Here we addressed multigenerational (grandparents and parents) effects of intergenerationally consistent versus changing diet experiences on cognition (learning novel prey) and fitness correlates (egg production) in the plant-inhabiting predatory mite *Amblyseius swirskii* Athias-Henriot (Acari, Phytoseiidae) (Fig. 1). Diet experiences made by plant-associated arthropods may be consistent or variable among generations, depending on food availability in their plant habitats changing with plant development, growth, and biotic and abiotic external influences or not. Accordingly, ancestors and offspring may experience diet-matching or diet-mismatching environments. *Amblyseius swirskii* is an ideal model animal to study the interaction between grandparental and parental effects on learning and its adaptive significance. *Amblyseius swirskii* are plant-inhabiting omnivorous predatory mites that feed on herbivorous mites and insects and plant-derived substances such as pollen[42–44]. Mixed animal and plant-based diets can be more or less favorable for life-history performance and population dynamics than monotypic diets, depending on the diet type combination[42,44,45]. In this study, the predators were offered three types of suitable diet also encountered in natural environments, that is, *Typha angustifolia* L. pollen, two-spotted spider mites *Tetranychus urticae* Koch (Acari, Tetranychidae), and western flower thrips *Frankliniella occidentalis* Pergande (Thysanoptera, Thripidae) (Fig. 1). Without causing any major differences, *Typha* pollen is somewhat more favorable as food than spider mites and both *Typha* pollen and spider mites are more favorable than thrips in terms of life-history performance such as immature survival and development and fecundity[43–45]. *Amblyseius swirskii* have been shown to possess well-advanced early life learning abilities, with early experience of a given prey type or its chemical cues having long-lasting effects on foraging behavior, such as enhanced prey recognition and capture and increased predation rates[46,47]. Thrips are more difficult-to-grasp and kill than are spider mites; accordingly, experiencing thrips early in life yields higher foraging benefits for the predators later in life than experiencing spider mites[47]. Early learning of prey is based on chemosensory cues and may be associative and non-associative[48,49]. The expression of early learning by offspring may be influenced by maternal effects. For example, maternal predation risk has been shown to affect offspring learning and behavior under predation risk[50].

It is untested for any animal how intergenerational change versus consistency between grandparents and parents affects the cognitive performance of offspring and the associated adaptive value in foraging contexts. Intergenerational diet variability could prime offspring for varying/unpredictable environments and thus allow offspring to better cope with novel diet/prey (e.g. by enhanced learning), whereas intergenerational consistency could prime them for a stable environment, compromising their ability to cope with novel prey (worsened learning performance). Further, we hypothesized that parental experience of moving prey should prime offspring (for example via epigenetic mechanisms)

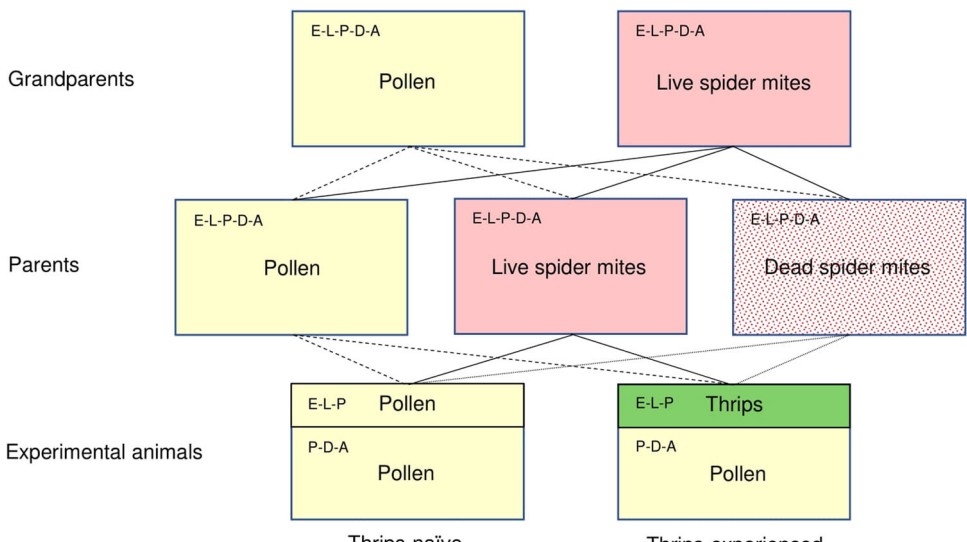

**Fig. 1 Experimental design.** Pre-experimental sequence of grandparental (F0), parental (F1), and personal (F2) diet experiences, representing a full 2 × 3 × 2 factorial design, resulting in 12 treatments of experimental animals (6 thrips-naïve and 6 thrips-experienced). Acronyms inside boxes are the life stages of the predators experiencing the diet indicated by the boxes; E for egg, L for larva, P for protonymph, D for deutonymph, and A for adult female. Mated experimental females that had experienced thrips early in life or not were then subjected to the behavioral assay.

to better cope with moving prey, whereas lacking parental experience of moving prey may compromise the offspring's ability of an adequate motor response to sensory input[51]. This may apply to foraging *A. swirskii*, because they rather go for mobile than immobile life stages of spider mites[52]. Transgenerational effects on offspring' foraging competence and skills are known from vertebrates when offspring observe foraging or predation or hunting by parents and thereby learn how to do it themselves[5]. However, we are not aware of any study that assessed the effects of parental experience of a highly critical aspect of foraging by an omnivorous predator, that is, moving versus non-moving prey (at similar nutritional contents), on key components of the offspring cognitive foraging phenotype, i.e. early life learning of novel prey affecting later searching, recognition, attack, and handling of moving prey.

## Results

Thrips-experienced predators successfully attacked and killed the 1st thrips earlier than did thrips-naïve predators, and both thrips-naïve and thrips-experienced progeny from spider mite-reared grandparents attacked the 1st thrips earlier than did progeny from pollen-reared grandparents (Fig. 2, Table 1, Supplementary Table 1); parental diet did not influence the latency to attack the 1st thrips (Fig. 2, Table 1). In a similar vein, thrips-experienced predators attacked/killed each of the three presented thrips individuals earlier (Fig. 3, Table 2, Supplementary Table 4) and killed more thrips within 5 h (Fig. 4, Table 2, Supplementary Table 5) than did thrips-naïve predators; grandparental spider mite diet shortened the attack latencies relative to grandparental pollen diet, which was especially the case in the combination with parental pollen diet (Fig. 3, Table 2). Progeny from purely spider mite-reared ancestors (parents and grandparents) attacked thrips later than did progeny from diet-switching ancestors (grandparents on spider mites, parents on pollen); similarly, progeny from purely pollen-reared ancestors (parents and grandparents) attacked thrips later than did progeny from diet-switching ancestors (grandparents on pollen, parents on spider mites) (Fig. 3, Table 2, Supplementary Table 4). Learned progeny killed more thrips within 24 h than did naïve progeny when they came from parents fed pollen or live spider mites. The reverse, naïve

progeny killing more thrips than did learned progeny, was the case when they came from parents fed dead spider mites (Fig. 5, Table 1, Supplementary Table 3). Thrips experience early in life increased egg production of progeny from parents reared on pollen or live spider mites but decreased egg production of progeny from parents reared on dead spider mites (Fig. 6, Table 1, Supplementary Table 2). Parental pollen diet increased egg production by progeny, which was especially true for diet-switching ancestors (grandparents on spider mites, parents on pollen). Within parental diets, the progeny of diet-switching ancestors (grandparents to parents; pollen to spider mites or vice versa) produced more eggs than the progeny of non-switching ancestors (Fig. 6, Table 1).

Body size of progeny (mm$^2$; mean ± SE) ranged from 0.0776 ± 0.0051 to 0.0788 ± 0.0069 and was neither influenced by grandparental (Wald $X_1^2 = 0.757$, $P = 0.384$) nor parental (Wald $X_2^2 = 0.792$, $P = 0.673$) diets (Supplementary Table 6 for estimated marginal means).

## Discussion

Our study documents how benign diet experiences in two ancestral generations (grandparents and parents) and their interactions combine with personal early life experiences to adaptively modulate the foraging phenotype of predatory mites. In all ancestral treatments, offspring remembered, as adults, the thrips cues experienced early in life. Shorter attack latencies on thrips several days after having made the experience, including two molting events and mating, points at the formation of long-term memory, as shown for *Drosophila*, by protein synthesis[53–56]. While the young predatory mites' basic learning ability was unaffected by the ancestral diets, the behavioral changes brought about by early life learning and their adaptive value varied with parental diet experience. Interaction of parental diet and early life learning was trait-dependent. Parental diet did not interact with early learning by offspring in the latency to attack prey (learned offspring were faster to attack thrips in all six combinations of grandparental and parental diet) but parental diet modulated the effects of early learning on the total number of thrips killed and the number of eggs produced. The ratio of the latter two traits is an indicator of prey profitability[57] and can be used as an indicator

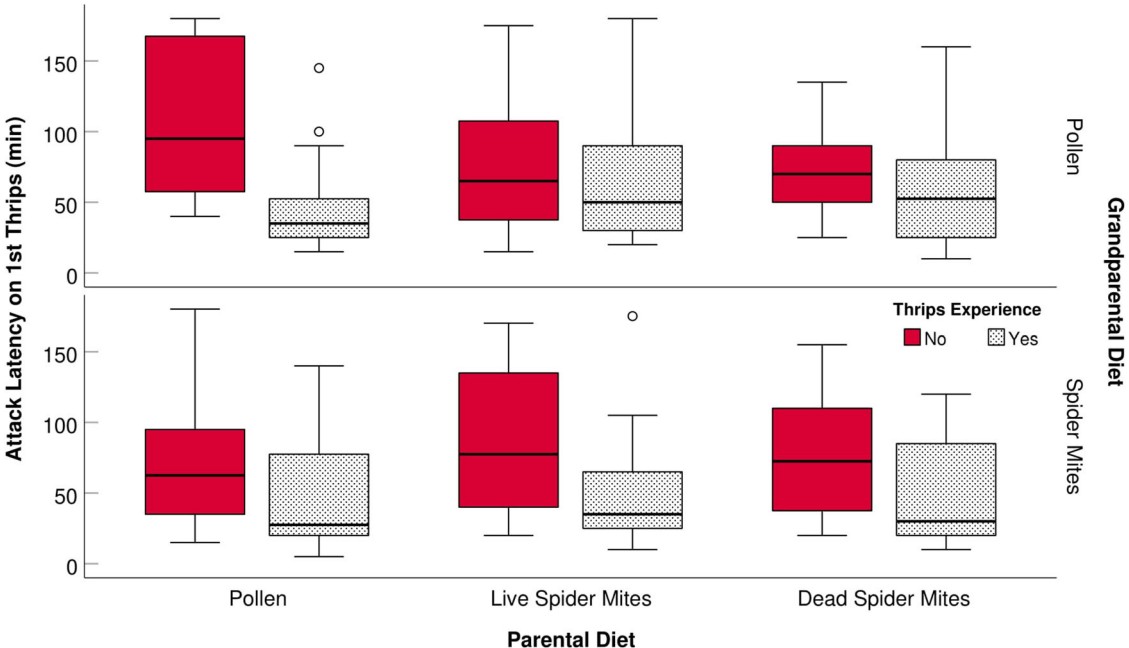

**Fig. 2 Latency to attack the first prey individual.** Latency (min; mean ± SE) of single mated *A. swirskii* females to attack and kill the 1st out of three 1st instar thrips, *F. occidentalis*, presented inside acrylic cages within 3 h. Predatory mite females had experienced thrips early in life or not (+ or −) and were the progeny of parents fed *Typha* pollen (P), life spider mites (L), or dead spider mites (D), and grandparents fed *Typha* pollen (P) or live spider mites (L). The numbers of replicates (*n*; independent experimental animals) within treatments (sequence in treatment acronyms is grandparental diet, parental diet, early life experience by progeny) are 20 (LP+), 20 (LP−), 18 (LL+), 14 (LL−), 17 (LD+), 20 (LD−), 15 (PP+), 16 (PP−), 21 (PL+), 16 (PL−), 14 (PD+), 15 (PD−). Boxes represent the 1st and 3rd quartile, the thick horizontal line inside boxes is the median, whiskers show the minimum and maximum values, symbols depict the outliers. Grandparental diet and thrips experience had significant effects (GLM, *P* < 0.05; see Table 1 for complete statistical results).

**Table 1 Generalized linear models (GLMs) on the influence of grandparental diet (pollen or spider mites), parental diet (pollen, live spider mites, dead spider mites) and thrips experience early in life (yes, no) on the attack latency on the 1st thrips (lg transformed; normal distribution, identity link function) and the number of thrips killed and eggs produced per female within 24 h (Poisson distribution, log-linear link function). Data are visualized in Figs. 2, 5, and 6; estimated marginal means are in Supplementary Tables 1–3.**

| Dependent and independent variables | Wald chi-square | df | P-value |
|---|---|---|---|
| *Attack latency on 1st thrips* | | | |
| (Intercept) | 6388.378 | 1 | <0.001 |
| Grandparental diet | 4.592 | 1 | 0.032 |
| Parental diet | 0.488 | 2 | 0.783 |
| Thrips experience | 25.71 | 1 | <0.001 |
| *Eggs per female per 24 h* | | | |
| (Intercept) | 52.668 | 1 | <0.001 |
| Grandparental diet | 4.032 | 1 | 0.045 |
| Parental diet | 40.166 | 2 | <0.001 |
| Thrips experience | 0.299 | 1 | 0.585 |
| Grandparental diet x Parental diet | 11.956 | 2 | 0.003 |
| Parental diet x Thrips experience | 11.234 | 2 | 0.004 |
| *Killed thrips per female per 24 h* | | | |
| (Intercept) | 4692.670 | 1 | <0.001 |
| Grandparental diet | 2.403 | 1 | 0.121 |
| Parental diet | 2.070 | 2 | 0.355 |
| Thrips experience | 0.358 | 1 | 0.550 |
| Parental diet x Thrips experience | 6.057 | 2 | 0.048 |

of the adaptive value of learning, i.e. the ability of foraging predatory mites to maximize the net energy gain by learning[47,48]. Further, our study suggests trait-dependency of grandparental and parental effects. Across all dependent variables (traits measured), grandparental and parental effects had about equal weight (weight inferred from the number of significant effects produced as main or interacting independent variables), but their relative weights differed within single traits and partially interacted. Interaction of grandparental and parental experiences points at updating of information passed on to the next generation with each generation[7,9,14]. Grandparental diet did not influence the difference in performance of learned and naïve mites, which was true for all traits measured, as indicated by the non-significant interaction effects of learning and grandparental diet. However, independent of learning, grandparental diet had significant effects on the attack latencies on all three thrips offered and the number of thrips killed.

Transgenerational effects spanning across multiple generations in benign environments have been rarely scrutinized but the available studies suggest enhanced offspring performance in matching ancestral and offspring environments, i.e. multigenerational anticipatory effects (Yin et al. 2019[9] for review). Our study shows that multigenerational effects may even persist and exert latent effects if the experience changes from one generation to the next, as evident from the grandparental effects occurring in treatments with intergenerational diet switch. Moreover, our study suggests that transgenerational effects may also be adaptive in non-matching offspring and ancestor environments. Partially, all offspring environments differed from ancestor environments because offspring were presented with novel prey (thrips), i.e. prey that was not experienced by their ancestors. Grandparental

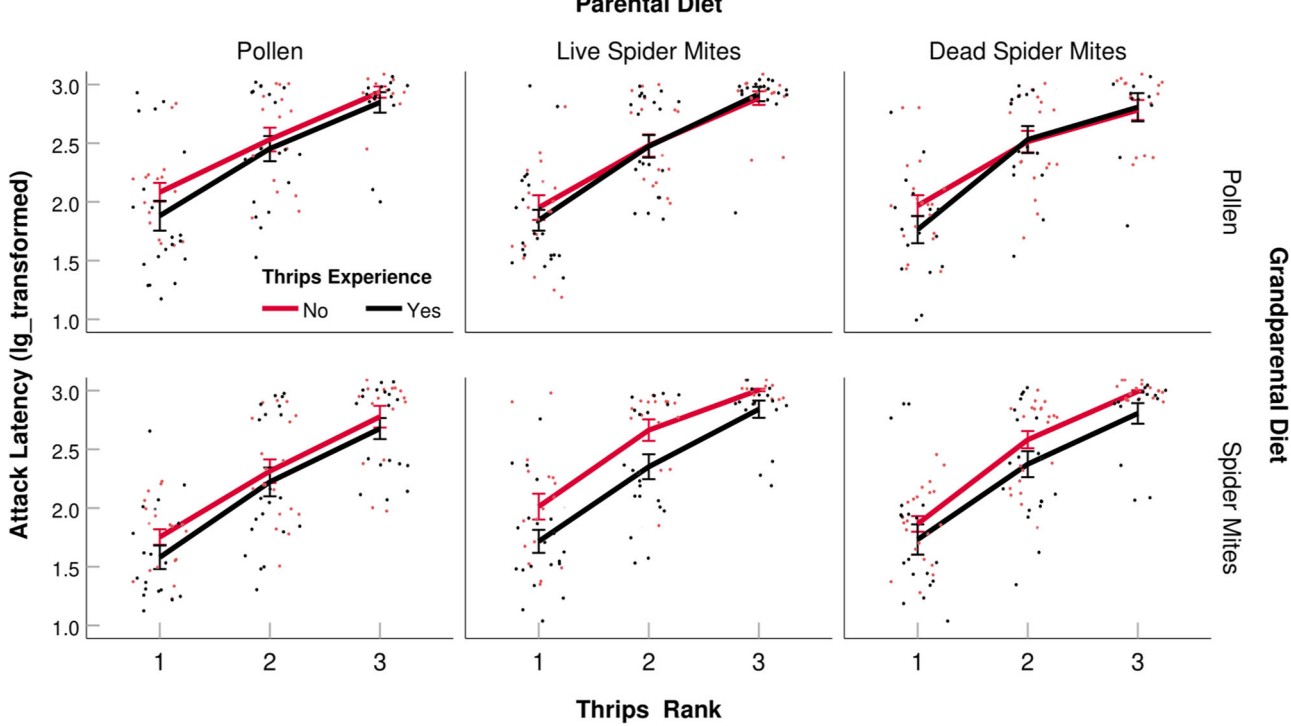

**Fig. 3 Latency to attack all three prey individuals.** Attack latency (min, lg_transformed; mean ± SE and individual data points) on the 1st, 2nd and 3rd out of three 1st instar thrips, *F. occidentalis*, presented to single mated *A. swirskii* females inside acrylic cages within 24 h. Predatory mite females had experienced thrips early in life or not (+ or −) and were the progeny of parents fed *Typha* pollen (P), life spider mites (L), or dead spider mites (D), and grandparents fed *Typha* pollen (P) or live spider mites (L). The numbers of replicates (*n*; independent experimental animals) within treatments (sequence in treatment acronyms is grandparental diet, parental diet, early life experience by progeny) are 21 (LP+), 21 (LP−), 21 (LL+), 18 (LL−), 20 (LD+), 22 (LD−), 20 (PP+), 20 (PP−), 23 (PL+), 20 (PL−), 16 (PD+), 19 (PD−). Grandparental diet, thrips experience, and their interaction had significant effects (GEE, *P* < 0.01; see Table 2 for complete statistical results).

**Table 2 Generalized estimating equations (GEEs) on the influence of grandparental diet (pollen or spider mites), parental diet (pollen, live spider mites, dead spider mites), and thrips experience early in life (yes, no) on the attack latency on all three thrips presented (lg transformed; normal distribution, identity link function) and the number of thrips killed within 1, 3, and 5 h (Poisson distribution, log-linear link function). M-dependent working correlation structure to account for the within-subject dependency of thrips' rank in the sequence and time, respectively. Data are visualized in Figs. 3, 4; estimated marginal means are in Supplementary Tables 4, 5.**

| Dependent and independent variables | Wald chi-square | df | P-value |
|---|---|---|---|
| *Attack latency on all three thrips* | | | |
| (Intercept) | 29356.993 | 1 | <0.001 |
| Grandparental diet | 7.055 | 1 | 0.008 |
| Parental diet | 4.67 | 2 | 0.097 |
| Thrips experience | 22.144 | 1 | <0.001 |
| Grandparental diet x Parental diet | 17.901 | 2 | <0.001 |
| *Number of thrips killed over time* | | | |
| (Intercept) | 0.319 | 1 | 0.572 |
| Grandparental diet | 1.689 | 1 | 0.194 |
| Parental diet | 0.647 | 2 | 0.724 |
| Thrips experience | 6.189 | 1 | 0.013 |

but not parental diet had an influence on latency to attack the 1st thrips and grandparental diet interacted with parental diet in attack latency on all three thrips offered and the number of eggs produced. Possible reasons why offspring emerging from the combination of pollen-fed parents and spider mite-fed grandparents were especially good in optimizing energy gain from thrips prey include being well nourished by the mixed (polytypic) diets of their ancestors[44] and/or a partially matching diet between parents and offspring, representing anticipatory parental effects[6,9]. However, since all offspring were fed pollen during immature development, intergenerational diet mixing was also the case for offspring from purely spider mite-fed ancestors and from ancestors switching diets between generations from pollen (grandparents) to spider mites (parents). An alternative or additional explanation for why offspring from spider mite-fed grandparents and pollen-fed parents performed the best in every trait across all treatments—they were the fastest, killed the most thrips, and produced the most eggs (both naïve and learned)—is that they might have had a weaker parentally caused predisposition for spider mite prey than offspring from spider mite-fed parents (assuming that the weight of such priming is weaker if stemming from grandparental than parental experiences). Due to cognitive trade-offs, a pronounced transgenerational effect on offspring' predisposition for a given prey type (spider mites) may have come at the cost of their ability to deal with a novel prey type (thrips)[46].

Proximately, we assume transgenerational effects that differ between grandparental and/or parental treatments irrespective of

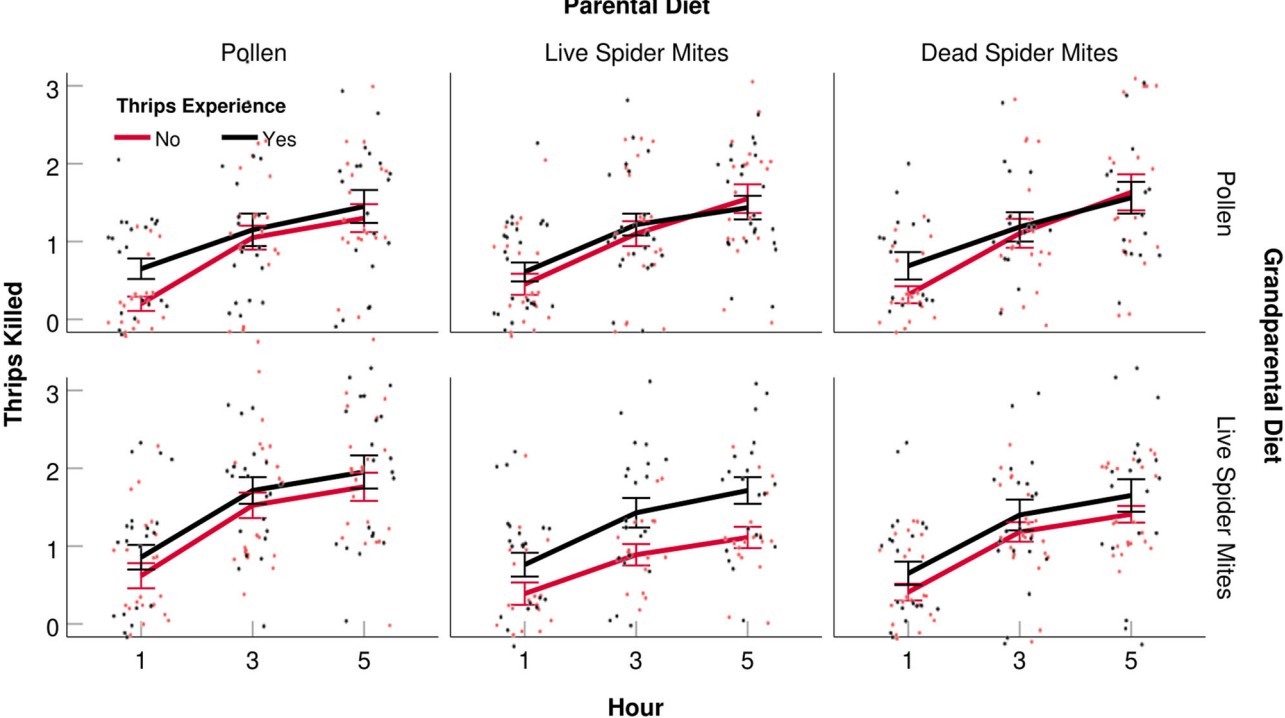

**Fig. 4 Predation rate over time.** Number (mean ± SE and individual datapoints) of 1st instar thrips, *F. occidentalis*, killed by single mated *A. swirskii* females inside acrylic cages within 1, 3, and 5 h. Predatory mite females had experienced thrips early in life or not (+ or −) and were the progeny of parents fed *Typha* pollen (P), life spider mites (L), or dead spider mites (D), and grandparents fed *Typha* pollen (P) or live spider mites (L). The numbers of replicates (*n*; independent experimental animals) within treatments (sequence in treatment acronyms is grandparental diet, parental diet, early life experience by progeny) are 21 (LP+), 20 (LP−), 21 (LL+), 18 (LL−), 20 (LD+), 22 (LD−), 20 (PP+), 19 (PP−), 23 (PL+), 20 (PL−), 16 (PD+), 19 (PD−). Thrips experience had a significant effect (GEE, *P* < 0.05; see Table 2 for complete statistical results).

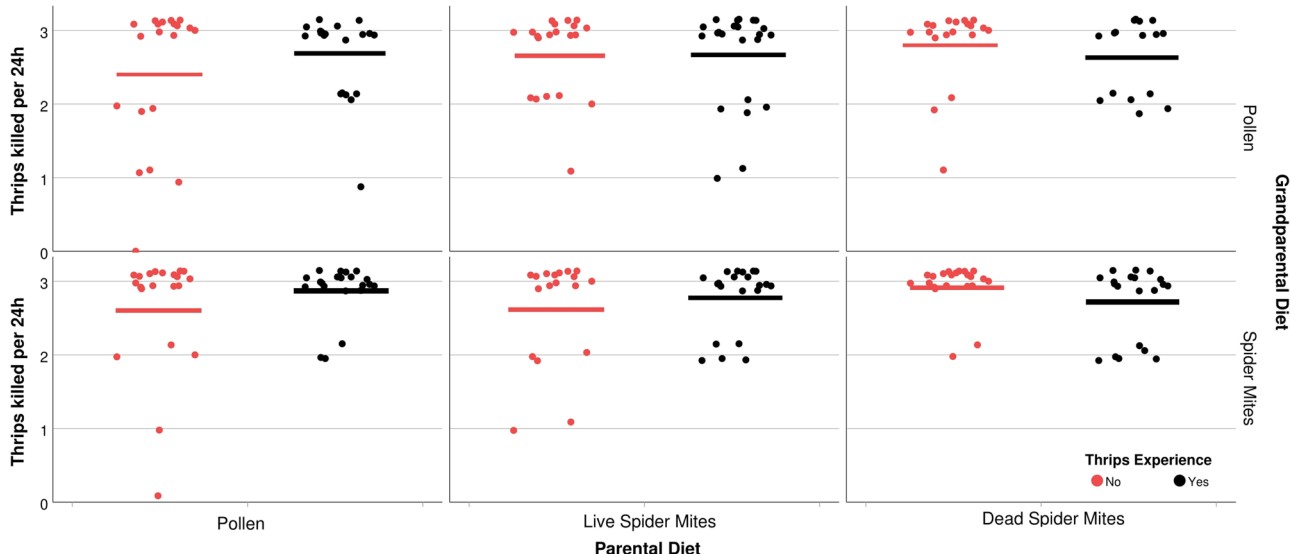

**Fig. 5 Total predation rate.** Number of thrips killed within 24 h (dots represent individual experimental animals; short horizontal bars represent the means) by mated *A. swirskii* females, kept singly inside acrylic cages and presented three 1st instar thrips, *F. occidentalis*. Predatory mite females had experienced thrips early in life or not (+ or −) and were the progeny of parents fed *Typha* pollen (P), life spider mites (L), or dead spider mites (D), and grandparents fed *Typha* pollen (P) or live spider mites (L). The numbers of replicates (*n*; independent experimental animals) within treatments (sequence in treatment acronyms is grandparental diet, parental diet, early life experience by progeny) are 21 (LP+), 21 (LP−), 21 (LL+), 18 (LL−), 20 (LD+), 22 (LD−), 20 (PP+), 20 (PP−), 23 (PL+), 20 (PL−), 16 (PD+), 19 (PD−). The interaction between parental diet and thrips experience had a significant effect (GLM, *P* < 0.05; see Table 1 for complete statistical results).

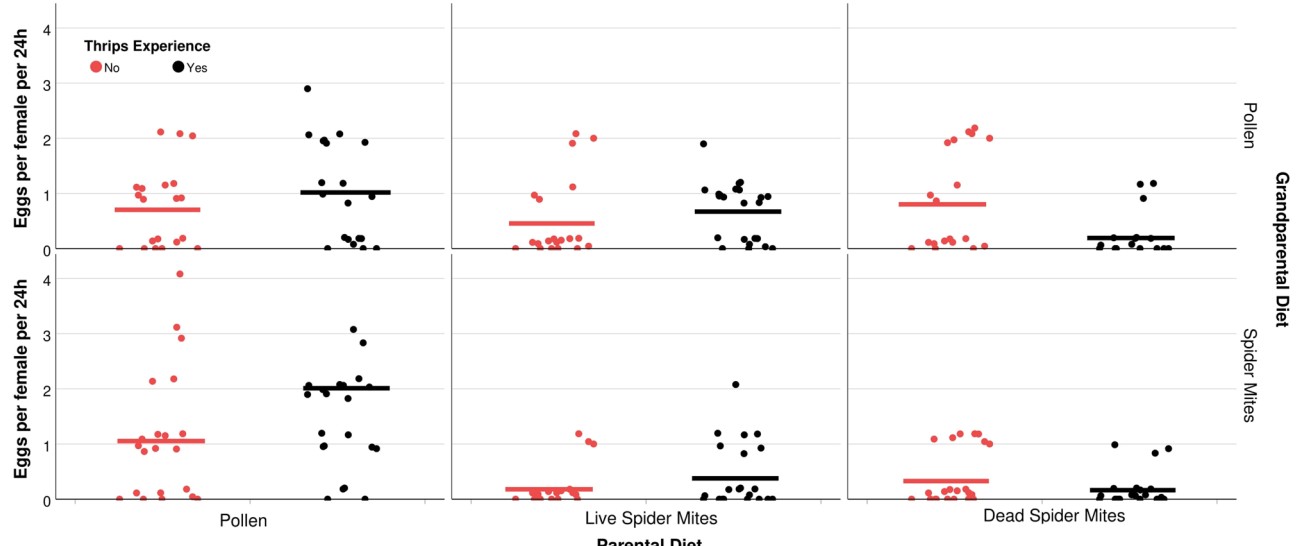

**Fig. 6 Oviposition rate.** Number of eggs produced within 24 h (dots represent individual experimental animals; short horizontal bars represent the means) by mated *A. swirskii* females, kept singly inside acrylic cages and presented three 1st instar thrips, *F. occidentalis*. Predatory mite females had experienced thrips early in life or not (+ or −) and were the progeny of parents fed *Typha* pollen (P), life spider mites (L), or dead spider mites (D), and grandparents fed *Typha* pollen (P) or live spider mites (L). The numbers of replicates (*n*; independent experimental animals) within treatments (sequence in treatment acronyms is grandparental diet, parental diet, early life experience by progeny) are 21 (LP+), 21 (LP−), 21 (LL+), 18 (LL−), 20 (LD+), 22 (LD−), 20 (PP+), 20 (PP−), 23 (PL+), 20 (PL−), 16 (PD+), 19 (PD−). Grandparental and parental diet, and the pairwise interactions of parental diet with thrips experience and with grandparental diet had significant effects (GLM, *P* < 0.05; see Table 1 for complete statistical results).

learning (thrips-experience) to be caused by nutritional effects (representing condition transfer or carry-over effects; e.g. Mendel and Schausberger 2011[58] for predatory mites), which may also include inheritance of metabolic state if a given nutrition causes metabolism-specific epigenetic marks[59]. Condition transfer is characterized by enhanced/reduced offspring condition following enhanced/reduced parental condition independent of offspring environmental quality, although the degree of enhancement/reduction may be context-dependent[60,61]. We consider the difference between learned and naïve offspring within a given treatment (grandparents/parents combination) to be cognitively caused because both learned and naïve predatory mites received the same nutrients from their ancestors and/or had been passed on the same metabolic state. Interactions of personal early life learning with parental diet may have been caused by the transmission of different nutrients affecting the expression of learning, including the acuity to chemical cues of thrips, and/or by different information transfer. Differences in learning between offspring from parents fed live and dead spider mites (such as the effects on eggs and the total number of thrips killed) are cognitively caused because we assume that the nutrients obtained from parents fed live and freshly killed spider mites were basically the same[62,63]. All transgenerational effects may have involved or may have been mediated by epigenetic changes induced by diet-specific features.

A particularly striking result deserving further scrutiny is the observed difference in behavior between offspring from parents fed moving and non-moving (freshly killed) spider mites. One possible proximate explanation for why thrips-experienced progeny from parents fed non-moving spider mites produced fewer eggs than thrips-naïve progeny from the same type of parents is that early learning non-adaptively changed prey profitability. Although thrips-experienced predators were quicker in finding thrips, like in the other treatments, based on early learning of chemical cues[46,48,49], they performed worse in handling thrips than naïve predators did (possibly by only partially sucking out thrips or investing more energy in overwhelming and sucking them out), which was reverse to all other treatments. Thrips-

experienced offspring from parents experiencing non-moving prey captured fewer thrips in total and produced fewer eggs than naïve individuals did. Difficulties in handling thrips by offspring might have come from a mismatch between the foraging phenotype of parents transmitted to offspring and the offspring early environment, i.e. parents priming offspring for non-moving prey. Such offspring might have experienced encounters with moving thrips early in life as stressful and later killed fewer thrips and laid fewer eggs than naïve predators did. This may be considered a silver-spoon effect[64–66]. Silver-spoon effects commonly arise if individuals experiencing stress early in life are in poorer condition and perform worse later in life than individuals experiencing benign conditions[65,66]. Also, Yin et al. (2019)[9] meta-analysis revealed that the combination of stressed parents and personal stress experienced by offspring commonly decreases reproduction, which matches the case here because we assume that being exposed to non-moving spider mites is mildly stressful for predators that are adapted to forage for moving spider mites. We observed during the parental phase that predators exposed to non-moving spider mites searched more and needed much longer to find prey than those offered moving spider mites. Ultimately, transgenerational transmission of foraging phenotypes for a given prey type, here moving or non-moving, may evolve to prepare offspring for the type of prey experienced by their parents and should be adaptive if prey types are matching between parental and offspring environments[7,12]. This was not the case in the parental treatment of non-moving prey. Offspring were parentally prepared for an environment with non-moving prey but encountered moving prey early in life, which lead to non-adaptive behaviors[67] by disassociating sensory input and motor response[28].

Overall, our study shows how benign diet experiences in multiple ancestor generations combined with personal early life experiences may produce adaptive behaviors in predators facing novel prey. We provide insights into transgenerational plasticity caused by persistent versus varying conditions in multiple ancestor generations and show that transgenerational effects may

also be adaptive in non-matching ancestor and offspring environments. These findings widen our understanding of the adaptive value of transgenerational plasticity and provide a key example to the pertinent research, which traditionally focuses on maternal effects caused by traumatic events such as predation risk, poor social conditions, or diet stress, and primarily assumes parental effects to be adaptive in matching environments.

## Methods

**Animal origins and rearing.** Specimens of *A. swirskii* (>200 individuals) used as founders of our laboratory population were obtained from Biohelp (Vienna, AT) and came originally from a population founded by specimens collected in the Near East and maintained by Koppert B.V. (Berkel en Rodenrijs, NL). Individuals from this population have been previously shown to perform well in learning tasks[48]. In the laboratory, the predatory mites were housed on black acrylic tile platforms (140 × 140 mm), resting on a foam block (140 × 140 × 50 mm) soaked in water and with moist tissue paper wrapped around all edges (subsequently called rearing arenas), with the foam block placed inside plastic boxes (200 × 200 × 60 mm) half-filled with water. Each rearing arena had five oviposition shelters, consisting of a few cotton wool fibers under coverslips (10 × 10 mm). The *A. swirskii* colonies were provided twice a week with a mixed diet comprised of cattail *T. angustifolia* pollen (Nutrimite, Biobest, BE) and two-spotted spider mites *T. urticae* (hereafter TSSM). Mixed TSSM life stages were brushed from infested detached common bean, *Phaseolus vulgaris* L. var. Maxi (Fabaceae), leaves using a Mite Brushing Machine (BioQuip, CA, USA) onto a circular glass plate and from there onto the rearing arenas. Rearing conditions provided for environmental heterogeneity and naturally encountered diet types and were thus close to what the predators experience in the field, i.e. pollen grains and spider mites on leaf surfaces.

The laboratory population of TSSM (red form) was founded with specimens sampled on tomato in Vienna (1020) and reared in the laboratory on whole bean plants *P. vulgaris*. Both *A. swirskii* and TSSM colonies were maintained in an air-conditioned room at 23 ± 1 °C and a 16:8 L:D photoperiod maintained by LED grow lights (SANlight FLEX 20). Founders of the laboratory population of western flower thrips *F. occidentalis* were obtained from D. Vangansbeke (Biobest, BE). In the laboratory, thrips were housed in glass or translucent plastic jars (500 and 750 mL) with a mesh cover for ventilation, kept at room temperature (21–23 °C) and ambient light. Fresh green bean pods were added to the jars 1 to 2 times a week, and *Typha* pollen was added once a week.

**Pre-experimental procedures.** The experiment followed a full 2 × 3 × 2 factorial design with two grandparental treatments, three parental treatments, and two personal early life experience treatments (Fig. 1).

*Grandparental treatments.* From the *A. swirskii* stock population, ~100 adult females were separated into two different lines, housed on arenas and conditions same as the stock population described above, except for diet provisioning. One line was fed only *Typha* pollen (PO), and the second line was fed only two-spotted spider mites (TSSM). Predatory mites were maintained on these two separate diets for several weeks before experiments to allow enough time for individuals to be used as grandparents to experience their respective diet from egg hatch onwards. Life history parameters such as survival and reproduction are similar on those two diets[43,44,68] resulting in similar densities on the rearing arenas.

From these two lines, ~20 gravid females, to be considered the grandparents of the experimental animals, were removed and placed on two separate leaf arenas (Fig. 1). Each leaf arena consisted of a large detached bean leaf resting upside down on a foam block (60 × 60 × 50 mm) soaked with water, and surrounded by moist tissue paper wrapped around the edges of the leaf. The foam block was placed inside a plastic box (100 × 100 × 60 mm) half-filled with water. One leaf arena was furnished with pollen and the other was furnished with two-spotted spider mites (Fig. 1), corresponding to the same diet of the line they came from. Females were allowed to freely oviposit, and these eggs were used in the parental treatments. Leaf arenas were housed in an environmental chamber (Panasonic MLR-352H-PE; 23 ± 1 °C, 60 ± 5% relative humidity RH, and 16:8 L:D photoperiod).

*Parental treatments.* Predatory mite eggs from each grandparental treatment (PO or TSSM) were randomly assigned to one of three parental treatments: fed pollen only (PO), fed live spider mites (L), or fed dead spider mites (D) (Fig. 1). For treatment D, spider mites were killed by deep-freezing at −30 °C for 30 min and immediately used thereafter. Spider mites killed by deep freezing and then immediately offered to the predators were considered to be similar in nutrients to live spider mites[62,63]. The result was six different combinations of the diet of the grandparents and parents: (i) PO-PO, (ii) PO-L, (iii) PO-D, (iv) TSSM-PO, (v) TSSM-L, and (vi) TSSM-D (Fig. 1). A cohort of 20–25 eggs from each of the two grandparental treatments (PO or TSSM) was placed onto each of six tile arenas corresponding to the three parental treatments (PO, L, or D); each tile arena consisted of a black acrylic tile platform (60 × 60 mm) resting on a foam block soaked with water and with moist tissue paper wrapped around the edges. Food was replenished once per day. These predatory mites, to become the parents of the

experimental animals, were provided with food ad libitum according to their diet treatment, and were left to develop to adult, mate, and allowed to oviposit freely on their arenas. Parental tile arenas were housed in an environmental chamber (23 ± 1 °C, 60 ± 5% RH, and 16:8 L:D).

*Personal early life experience treatments.* All eggs, to become the experimental animals, were removed daily from each of the six parental arenas, and randomly assigned to one of two early life treatments: (i) exposed to pollen as larvae and early protonymphs (PO), to be considered thrips-naïve, or (ii) exposed to thrips as larvae and early protonymphs (FO), to be considered thrips-experienced. This led to twelve treatments of experimental animals (Fig. 1). Eggs were placed individually into a cylindrical acrylic cage (15 mm Ø × 3 mm high) closed with fine mesh at the bottom and a removable glass slide fixed by foldback (binder) clips on the upper side[68] and loaded with either pollen or three dead (killed immediately before by deep-freezing for 30 min) 1st to 2nd instar thrips nymphs. Killed thrips were used because live thrips tend to kill predatory mite eggs[69]. Predators in these cages were inspected daily under a stereo-microscope for hatching into larvae (after 1–2 days) and molting to protonymphs (after 2–3 days). As soon as the predator larva was observed, 1 to 2 live 1st instar thrips were added to the cage. Larvae of *A. swirskii* do not have to eat to molt to protonymph[43]. As soon as the protonymph was observed, it was taken out from the cage, using a fine moistened marten's hairbrush (0), and transferred to a tile arena (6 × 6 cm; all protonymphs were grouped on one arena per treatment), fed pollen ad libitum, and allowed to develop to adult and mate. Cages were placed upside down on a grid above moistened paper towels on a plastic tray housed in an environmental chamber (23 ± 1 °C, 60 ± 5% RH, and 16:8 L:D).

**Behavioral assay.** Mated females (mating status recognizable by the extended idiosoma) from each of the twelve treatments were individually transferred to acrylic cages (constructed as described above) that had been previously loaded with three live 1st instar thrips. Cages were monitored every 10–20 min over 180 min and again at 300 min after starting the behavioral assay to record the time and number of thrips killed by the predators, and after 24 h for killed thrips and the number of eggs laid by the predators. Latency to attack thrips (resulting in the killing of thrips) was determined by the time elapsed after presenting thrips until directly observing the successful attack (resulting in killing), or, if killed and sucked out thrips were found after 300 min and 24 h, by approximation of the nearest killing times (240 min if killed thrips was found after 300 min but not at 180 min; 870 min for one killed thrips, 680 and 1060 min for two killed thrips, and 585, 1070, and 1155 min for three killed thrips found after 24 h but not at 300 min). Cages were placed on a plastic tray housed in an environmental chamber (23 ± 1 °C, 60 ± 5% RH and 16:8 L:D) and periodically taken out for monitoring under a stereo-microscope (at ambient room temperature of 21–23 °C). Each treatment was replicated 16 to 23 times (20 PO-PO-naïve, 20 PO-PO-experienced, 20 PO-L-naïve, 23 PO-L-experienced, 19 PO-D-naïve, 16 PO-D-experienced; 21 TSSM-PO-naïve, 21 TSSM-PO-experienced, 18 TSSM-L-naïve, 21 TSSM-L-experienced, 22 TSSM-D-naïve, 20 TSSM-D-experienced). Each acrylic cage, thrips and predator individual were used only once; all twelve treatments were run in parallel.

**Body size measurements.** Since latency to attack, predation rates and number of eggs produced may scale with the body size of predatory mites[70,71] we checked for the influence of grandparental and parental diet on the body size of progeny. Mated females coming from all six combinations of grandparental and parental diet treatments and reared themselves on pollen were mounted in a drop of lactic acid on microscope slides and the microscope slides dried on a heating plate at 50 °C for 24 h for clearing the mites[72]. The maximal length and width of the dorsal shield, which is a suitable indicator of body size[73], were measured under a transmitted phase-contrast light microscope (Nikon Eclipse 200) at 150x magnification using a reticle, and the area of the dorsal shield was calculated using the formula of an ellipse ($a × b × \pi$); the dorsal shield of *A. swirskii* is geometrically best characterized by the shape of an ellipse. Ten to 22 replicate females of each of the six combinations of the two grandparental and three parental diet treatments were measured (19 PO-PO, 22 PO-L, 10 PO-D; 16 TSSM-PO, 21 TSSM-L, 18 TSSM-D).

**Statistics and reproducibility.** IBM SPSS Statistics ver. 28 (IBM Corp., Armonk, NY, USA) was used for all statistical analyses and Figs. 2–6. All analyses were two-tailed. Generalized linear models (GLM) were used to assess the influence of grandparental diet (pollen or TSSM), parental diet (pollen, live TSSM or dead TSSM), and early life thrips experience (yes or no) on the attack latency on the 1st thrips within the first 3 h (lg transformed before analysis to normalize the data; normal distribution, identity link), the number of thrips killed within 24 h (Poisson distribution, log-linear link) and the number of eggs laid within 24 h (Poisson distribution, log-linear link). GLM (normal distribution, identity link) was also used to analyze the influence of grandparental and parental diets on body size (area of the dorsal shield) of progeny. Generalized estimating equations (GEE) were used to assess the influence of grandparental diet, parental diet and early life thrips experience on the attack latency on all three thrips individuals within 24 h (lg transformed before analysis to normalize the data) and the number of thrips killed

within 1, 3, and 5 h (M-dependent working correlation structure to account for the within-subject dependency of the rank of thrips and time, respectively). In each analysis, we started with the full model and removed non-significant interaction terms until arriving at the most parsimonious model (>2 difference) based on the Bayesian Information Criterion (BIC) for GLMs and the Quasi Likelihood under Independence Model Criterion (QIC) for GEEs (Supplementary Note 1 and Supplementary Tables 1–6).

**Reporting summary**. Further information on research design is available in the Nature Research Reporting Summary linked to this article.

## Data availability

The raw data underlying the current study are available on Figshare (https://doi.org/10.6084/m9.figshare.19243869). All other data are available from the corresponding author upon reasonable request.

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

## Acknowledgements
The authors acknowledge financial support by the Austrian Science Fund (FWF; P 33787-B to PS).

## Author contributions
P.S. conceived the study, designed the experiment, acquired funding, analyzed the data, wrote the first draft, and revision of the manuscript; P.S. and D.R. conducted the experiment; D.R. contributed to writing and revision of the manuscript.

## Competing interests
The authors declare no competing interests.
