## [Peer Review File · Communications Biology]

Reviewers' comments:

Reviewer #1 (Remarks to the Author):

This study asks the question, do the experiences of ancestors (grandparents and parents) influence the behavior of offspring, in addition to their own early-life experiences? The study uses predatory mites and their foraging phenotypes to explore this question. The authors make the argument (and I believe they are correct) that previous studies have emphasized stressful experiences on the part of ancestors (such as nutritional stress), whereas they focus on diet variation that is "benign" for the mites. Studies that examine transgenerational effects of grandparents and parents, including when their experiences conflict, are rare, and I think this is a very interesting area of research in that it can help explain how epigenetic and other effects are precipitated through time (or not).

Overall the study is well designed; the sample sizes are sufficient. The study design is clearly outlined. The statistical analyses are appropriate (but see my comment below about post hoc tests). It was good to see that personal experience (with thrip predation) did influence successful predation on thrips later in life, which was a bit of a positive control in the experimental design.

There are limitations to what can be inferred from the study in that it did not look at underlying mechanisms, and the treatments by their nature differ in many variables, not only whether they are mobile or not.

Minor comments:

Lines 89-93: Some sentences are a bit long / complicated, for example this one.

Line 329-331: For this statement, is this something that has been observed in previous studies (in which these should be cited), and/or was this the pattern observed in the current study as well?

Figures: It would be good to run pairwise post hoc tests to determine significant differences between groups, then include symbols on figures that show significant differences between groups. Otherwise it is difficult to cross reference the results and interpret the figures. Including results of GLMM and other statistical tests in figure legends would also be helpful.

Reviewer #2 (Remarks to the Author):

Dear authors,

Please find below comments on the manuscript "Transgenerational effects of multiple ancestor diets combine with early-life learning to shape adaptive foraging phenotypes"

The authors aim to investigate the presence and influence of transgenerational diet experiences on the foraging phenotypes of predatory mites. More specifically they test whether grandparental and parental exposure to either pollen or spider mites (alive or dead) influence latency to attack thrips, number of thrips killed, and number of eggs laid. In addition, they test whether these diets influence body size, which can directly influence the above traits. The authors found evidence to suggest that parental and ancestral diets interact to alter foraging and fitness of the predatory mites.

Overall I did find this paper interesting but feel that a lot of it could be rephrased for clarity and ease of reading. In addition, I feel the statistical tables/methodology – in particular tables relating to model selection – are lacking a lot of information that could help with interpretation. Also, the test of body size is only really introduced during the results/methods and doesn't get discussed further in the context of what was found particularly as the authors highlight previous work suggesting that size may act as an important factor.

Comments:

→L36-38. It might be worth specifically mentioning/noting in general somewhere here or L44-45 that for an effect to be considered “transgenerational” it has to be detected in at least the F2 generation. With appropriate references e.g., O’Dea 2016 “The role of non-genetic inheritance in evolutionary rescue: epigenetic buffering, heritable bet hedging and epigenetic traps” or Perez 2019 “Intergenerational and transgenerational epigenetic inheritance in animals”. However, in general this sentence is quite longwinded, I would try and write it a tad more succinctly.

Introduction in general: A lot of these sentences start with the same “Transgenerational effects...”, it makes it sound like a list of facts and doesn’t flow that well. Instead, you might want to change some of these with “They”?

In addition, you test for the effects of body size but make no mention of this trait in your introduction.

L48-50. These are commonly referred to as inter-generational effects or, as written in text, parental effects. I would make the distinction between trans- and inter-generational effects clear.

L56-57: I would also make a note of asking whether or not they are indeed adaptive? “The jury is still out regarding the generality of adaptive ‘transgenerational’ effect” Sanchez-Tojar 2020.

L62-64: Perhaps give more information on what these effects are.

L67: “have focused”.

L74: “conditions with little stress” or something similar. Though I’m not sure if it’s necessary to repeat the definition of benign twice.

L76: Remove “and”.

L77: I would try and rewrite the end of the sentence or just remove it (“for this is the most common state”).

L90: Remove “just”.

L91: Replace “erasing” with “removal”.

L98-99: This sentence doesn’t flow very well from the previous sentence.

L100: The opening sentence was regarding transgenerational effects, but now you use multi-generational. I would stick with transgenerational for consistency.

L106: “diet-mismatching?”.

L177: Differences to fitness? I would be specific here. Perhaps reorganise the last part of this sentence regarding life history performance to before the comma.

L131 and elsewhere: “inter-generational” and “multigenerational”. Be consistent with hyphens or no hyphens.

L141: “Parental effects” similarly, the idea of 2 or more generations to be considered transgenerational.

Results: The body mass paragraph contains statistical results, but the first paragraph does not. You should include the relevant information such as effect sizes and p values for instance.

L155: Is there a distinction between attack and kill?

L163-166: This sentence is difficult to follow and very long. I realise the results are complex but

there must be a better way to explain these results so it's easier to follow.

L166: This sentence makes it sound like the early life thrips had increased egg production. I would rephrase.

L178 and elsewhere: I would check to see if "ancestral" fits better rather than "ancestor".

L180-181: I'm not sure what this sentence means.

L185: "by ancestral diets".

L185-191: These sentences are about early-life learning and parental diet experience, and in each sentence these two terms are repeated. I would try and make these sentence flow better.

L201-204: This sentence is very long and is difficult to understand. I would shorten and rephrase.

L207-208: But again this has been queried... "The jury is still out regarding the generality of adaptive 'transgenerational' effect" Sanchez-Tojar 2020.

L221: "were fed pollen..."

L222: "diets".

L280-281: "moving prey early in life" and "behaviors".

L359: How many eggs? All eggs? In addition, eggs were place into cages, how many cages?

L363: I would add details on sample sizes for each combination or add it to figure 1.

L390-391 and L404. Is there any reason for the range in replicates, also which number corresponds to which treatment?

Statistical analysis: I think that these models could be underspecified, although, it might just be that the methods are reasonably hard to follow when determining what should and shouldn't be accounted for i.e. with random effects. For instance, I'm not versed in SPSS but I assume L418 is regarding repeated measuring of the same individuals?

Lastly, why are both AIC and BIC used, I would only use one as often they don't give the same answer...

Figure 3: What is meant be Thrips_Rank. Instar?

Tables 1-2. These tables need far more information. Such as effect sizes and error rather than just a focus on significance. In addition, there doesn't seem to be information regarding model selection?

Response to the referees' comments (our response is highlighted in red; line numbers refer to the version with marked changes)

Reviewer #1 (Remarks to the Author):

This study asks the question, do the experiences of ancestors (grandparents and parents) influence the behavior of offspring, in addition to their own early-life experiences? The study uses predatory mites and their foraging phenotypes to explore this question. The authors make the argument (and I believe they are correct) that previous studies have emphasized stressful experiences on the part of ancestors (such as nutritional stress), whereas they focus on diet variation that is "benign" for the mites. Studies that examine transgenerational effects of grandparents and parents, including when their experiences conflict, are rare, and I think this is a very interesting area of research in that it can help explain how epigenetic and other effects are precipitated through time (or not).

Overall the study is well designed; the sample sizes are sufficient. The study design is clearly outlined. The statistical analyses are appropriate (but see my comment below about post hoc tests). It was good to see that personal experience (with thrip predation) did influence successful predation on thrips later in life, which was a bit of a positive control in the experimental design.

There are limitations to what can be inferred from the study in that it did not look at underlying mechanisms, and the treatments by their nature differ in many variables, not only whether they are mobile or not.

This study set out to test for effects spanning across multiple generations, whether they exist and, if yes, whether they interact, what they cause and how they combine with personal early life experiences. Based on the results of this study, next steps are scrutinizing and disentangling possible proximate mechanisms. These tests were beyond the scope of this study.

We agree that the three diet treatments differ in various aspects but the live and dead spider mite diets differ primarily in activity and movement of the prey individuals, assuming that the nutrient contents of live and dead prey were basically the same (this assumption is based on killing by shock-freezing and immediately thereafter presenting dead prey to the predators not altering the nutrient content as compared to live prey) - see lines 257 to 260.

Minor comments:

Lines 89-93: Some sentences are a bit long / complicated, for example this one.

Changed (lines 91 to 97)

Line 329-331: For this statement, is this something that has been observed in previous studies (in which these should be cited), and/or was this the pattern observed in the current study as well?

References included (lines 340 to 342)

Figures: It would be good to run pairwise post hoc tests to determine significant differences between groups, then include symbols on figures that show significant differences between groups. Otherwise it is difficult to cross references the results and interpret the figures. Including results of GLMM and other statistical tests in figure legends would also be helpful.

Post-hoc tests are not meaningful in our analyses because of significant interaction terms. We now also report the significant effects in the figure captions and refer to tables 1 and 2 for the full results of statistical analysis.

Reviewer #2 (Remarks to the Author):

Dear authors,

Please find below comments on the manuscript “Transgenerational effects of multiple ancestor diets combine with early-life learning to shape adaptive foraging phenotypes”

The authors aim to investigate the presence and influence of transgenerational diet experiences on the foraging phenotypes of predatory mites. More specifically they test whether grandparental and parental exposure to either pollen or spider mites (alive or dead) influence latency to attack thrips, number of thrips killed, and number of eggs laid. In addition, they test whether these diets influence body size, which can directly influence the above traits. The authors found evidence to suggest that parental and ancestral diets interact to alter foraging and fitness of the predatory mites.

Overall I did find this paper interesting but feel that a lot of it could be rephrased for clarity and ease of reading. In addition, I feel the statistical tables/methodology – in particular tables relating to model selection - are lacking a lot of information that could help with interpretation. Also, the test of body size is only really introduced during the results/methods and doesn't get discussed further in the context of what was found particularly as the authors highlight previous work suggesting that size may act as an important factor.

The predator-prey body size ratio is a potential issue in many predator-prey interactions but we did not have an *a priori* hypothesis or expectation that predator body size should differ among treatments and/or influence the foraging traits examined. To make sure that in our experiments and treatments body size was not a confounding factor for differences ascribed to experience or nutritional state, we checked whether the different diets induced size differences.

Comments:

–L36-38. It might be worth specifically mentioning/noting in general somewhere here or L44-45 that for an effect to be considered “transgenerational” it has to be detected in at least the F2 generation. With appropriate references e.g., O’Dea 2016 “The role of non-genetic inheritance in evolutionary rescue: epigenetic buffering, heritable bet hedging and epigenetic traps” or Perez 2019 “Intergenerational and transgenerational epigenetic inheritance in animals”. However, in general this sentence is quite longwinded, I would try and write it a tad more succinctly.

Rephrased (lines 37 to 39). In general, we prefer to stick with the broader definition of transgenerational effects, as in Yin et al. (2021) and numerous other pertinent references, that is, non-genetic transfer of information from one generation to the next or across multiple generations. This broader definition also includes parental effects. We are aware of the terminological specifics between transgenerational and intergenerational effects in epigenetics (Heard and Martienssen 2014, O’Dea et al. 2016, Perez and Lehner 2019) but refrain from applying this epigenetic-centered terminology because the effects observed in our study may be mediated by a number of mechanisms, instead or in addition to epigenetic changes, or combinations thereof (see paragraph in the discussion in lines 243 to 261).

Introduction in general: A lot of these sentences start with the same “Transgenerational effects...”, it makes it sound like a list of facts and doesn’t flow that well. Instead, you might want to change some of these with “They”?

Changed accordingly (lines 36 to 58)

In addition, you test for the effects of body size but make no mention of this trait in your introduction.

The predator-prey body size ratio is a potential issue in many predator-prey interactions but we did not have an *a priori* hypothesis or expectation that in our experiments predator body size should differ among treatments and/or influence the foraging traits tested. To make sure that body size was not a confounding factor in our study, we checked whether the grandparental and/or parental diets induced size differences.

L48-50. These are commonly referred to as inter-generational effects or, as written in text, parental effects. I would make the distinction between trans- and inter-generational effects clear.

See above response to the first comment.

L56-57: I would also make a note of asking whether or not they are indeed adaptive? “The jury is still out regarding the generality of adaptive ‘transgenerational’ effect” Sanchez-Tojar 2020.

We are aware of the comment by Sanchez-Tojar et al. (2020) and also know the valid reply by Zhang et al. (2020) to this comment. We consider Yin et al.’s (2019) review a valid source of information about the adaptive value of transgenerational effects.

L62-64: Perhaps give more information on what these effects are.

Done (lines 61 to 66)

L67: “have focused”.

Done (line 68)

L74: “conditions with little stress” or something similar. Though I’m not sure if it’s necessary to repeat the definition of benign twice.

We use benign for conditions without any stress and little stressful for conditions with little stress.

L76: Remove “and”.

Done (line 77)

L77: I would try and rewrite the end of the sentence or just remove it (“for this is the most common state”).

Rephrased (lines 77 to 79)

L90: Remove “just”.

Done (line 92)

L91: Replace “erasing” with “removal”.

Done (line 96)

L98-99: This sentence doesn't flow very well from the previous sentence.

Rephrased (lines 102 to 104)

L100: The opening sentence was regarding transgenerational effects, but now you use multi-generational. I would stick with transgenerational for consistency.

Not changed. Multi-generational has a different meaning than transgenerational

L106: "diet-mismatching?".

Changed (line 112)

L177: Differences to fitness? I would be specific here. Perhaps reorganise the last part of this sentence regarding life history performance to before the comma.

We did not find the relevant section in line 177? We assume this comment refers to line 105, where we now added egg production in parenthesis after fitness correlates (lines 106, 107)

L131 and elsewhere: "inter-generational" and "multigenerational". Be consistent with hyphens or no hyphens.

Changed throughout the manuscript

L141: "Parental effects" similarly, the idea of 2 or more generations to be considered transgenerational.

See above response to the first comment.

Results: The body mass paragraph contains statistical results, but the first paragraph does not. You should include the relevant information such as effect sizes and p values for instance.

The statistical information for all results reported in the first paragraph is given in tables and figures, which is not the case for body size, which information is just given in text.

L155: Is there a distinction between attack and kill?

Yes, attacks may be successful or not, kills are successful attacks (see lines 394 to 399)

L163-166: This sentence is difficult to follow and very long. I realise the results are complex but there must be a better way to explain these results so it's easier to follow.

Split into two sentences and rephrased (lines 168 to 171).

L166: This sentence makes it sound like the early life thrips had increased egg production. I would rephrase.

Rephrased (lines 174 to 176)

L178 and elsewhere: I would check to see if "ancestral" fits better rather than "ancestor".

Checked and changed throughout the manuscript

L180-181: I'm not sure what this sentence means.

Rephrased (lines 188 to 190)

L185: “by ancestral diets”.

Changed (lines 193 to 196)

L185-191: These sentences are about early-life learning and parental diet experience, and in each sentence these two terms are repeated. I would try and make these sentence flow better.

Rephrased (lines 193 to 200)

L201-204: This sentence is very long and is difficult to understand. I would shorten and rephrase.

Rephrased (lines 213 to 215)

L207-208: But again this has been queried... “The jury is still out regarding the generality of adaptive ‘transgenerational’ effect” Sanchez-Tojar 2020.

We are aware of the comment by Sanchez-Tojar et al. (2020) on Yin et al. (2019) and the reply by Zhang et al. (2020) to this comment. In light of this discussion, we consider Yin et al.’s (2019) review a valid source of information about the adaptive value of transgenerational effects.

L221: “were fed pollen...”

Changed (line 232)

L222: “diets”.

Changed (line 234)

L280-281: “moving prey early in life” and “behaviors”.

Changed (lines 290 to 293)

L359: How many eggs? All eggs? In addition, eggs were place into cages, how many cages?

All eggs; the exact number of replicates (with one egg per cage) is given in lines 370 to 373

L363: I would add details on sample sizes for each combination or add it to figure 1.

Done, see lines 402 to 406

L390-391 and L404. Is there any reason for the range in replicates, also which number corresponds to which treatment?

Exact number of replicates inserted for each treatment (lines 402 to 406); the number of replicates is not exactly the same for all treatments because some replicates were lost during the experiment, due to handling errors or other unnatural causes, or the parents produced fewer eggs or fewer females in a given treatment.

Statistical analysis: I think that these models could be underspecified, although, it might just be that the methods are reasonably hard to follow when determining what should and shouldn’t be accounted for i.e. with random effects. For instance, I’m not versed in SPSS but I assume L418 is regarding repeated measuring of the same individuals?

The statistical models are accounting for all independent factors differing among treatments; your assumption about GEE is correct, generalized estimating equations (GEE) are perfectly suitable models to account for repeated measurements and interdependency of observations.

Lastly, why are both AIC and BIC used, I would only use one as often they don't give the same answer...

In fact, BIC is more stringent than AIC in penalizing non-parsimony, so in all cases where the difference in AIC was larger than 2, the difference in BIC was also larger than 2 but not vice versa. We now just write ...based on BIC...

Figure 3: What is meant by Thrips_Rank. Instar?

Thrips rank refers to the sequence of attack within the three thrips individuals offered as prey. Rank 1 is the first thrips attacked, rank 2 is the second thrips attacked, rank 3 is the third thrips attacked. For clarification, we added an explanation to the caption of figure 3.

Tables 1-2. These tables need far more information. Such as effect sizes and error rather than just a focus on significance. In addition, there doesn't seem to be information regarding model selection?

The effect sizes and standard errors are visualized in figures 2 to 6. Tables 1 and 2 accompany the figures to report the results of statistical analyses. Reporting the statistical values inside text would compromise readability. We now provide more statistical information in the figure captions and additionally refer to the tables for complete statistical results. Model selection is explained in the paragraph on statistics and reproducibility.

REVIEWERS' COMMENTS:

Reviewer #1 (Remarks to the Author):

I do not have any additional comments related to the manuscript.

Reviewer #2 (Remarks to the Author):

Dear authors,

Please find below my comments on the manuscript "Transgenerational effects of grandparental and parental diets combine with early-life learning to shape adaptive foraging phenotypes".

The authors have responded and edited the manuscript according to the previous round of reviews and I do think the manuscript has improved. However, I still think a few parts of the paper are quite difficult to follow, mainly owing to complicated sentence structure. In addition, as mentioned in the last round of reviews, I think the tables need to contain the full outputs with associated estimates and standard errors, with an additional table for body size. Lastly, the need for increased detail and tables relating to model selection – ie which terms were dropped and comparison of models with BIC/QIC.

Comments:

↴

L1-2: This sentence is difficult to read. Perhaps rewrite to:

"Transgenerational effects are non-genetic environmental influences that are passed from one generation to the next and typically cause phenotypic variation without a change in the sequence of DNA"... or something similar?

L47: I would rephrase to "maternal and paternal, or more generally, parental effects).

L48: Replace "for" with "in".

L49: Replace the first "and" with "within".

L53: Add a comma after "effects" and after "circumstances".

L61: Insert a bracket after "fruit flies".

L65: I realise this was mentioned in the previous round of review – but "little stressful" doesn't sound right. "Conditions with little stress" seems more appropriate.

L80: Change to "rather than simply parental information" or something similar.

L86: Add a comma after "generations".

L93-96: This sentence is rather long and a bit difficult to read. Rewrite.

L115-116: This last line doesn't flow well, I'd see if you could link it better to the previous sentence. Similarly on L126-127.

L160-162: I would remove the statistical text from this paragraph and create a new table with all of the relevant information and simply reference that, similar to above for consistency.

L168: Remove the third comma.

L170: You might want to expand upon why it points to protein synthesis by specifically referring to the referenced papers in *Drosophila*, otherwise this point kind of appears out of nowhere.

-iL345: "thrips naïve"?

Tables – I noticed in the response to reviewers that the notion of presenting effect sizes in tables was not considered. For ease of reading, and not having to work out the effect from the graph, it would better to have the effect size and standard error in each table for each model. Particularly as the estimates from models will often not be similar to the raw data in light of random/fixed effects.

The full model outputs with appropriate model selection (which is mentioned in the methods but is not then available to review in the paper or in the SM – unless I've missed it) should then be placed in the supplementary.

Figure 3: Replace Thrips_Rank with Thrips Rank.

Response to the reviewers (line numbers refer to the version with marked changes)

Reviewer #1 (Remarks to the Author):

I do not have any additional comments related to the manuscript.

Reviewer #2 (Remarks to the Author):

Dear authors,

Please find below my comments on the manuscript “Transgenerational effects of grandparental and parental diets combine with early-life learning to shape adaptive foraging phenotypes”.

The authors have responded and edited the manuscript according to the previous round of reviews and I do think the manuscript has improved. However, I still think a few parts of the paper are quite difficult to follow, mainly owing to complicated sentence structure. In addition, as mentioned in the last round of reviews, I think the tables need to contain the full outputs with associated estimates and standard errors, with an additional table for body size. Lastly, the need for increased detail and tables relating to model selection – ie which terms were dropped and comparison of models with BIC/QIC.

We rephrased several sections of the manuscript to ease readability. The requested details of model selection and effect sizes are now provided in the supplementary material.

Comments:

–

L1-2: This sentence is difficult to read. Perhaps rewrite to:

“Transgenerational effects are non-genetic environmental influences that are passed from one generation to the next and typically cause phenotypic variation without a change in the sequence of DNA”... or something similar?

Rephrased (L 36 to 38)

L47: I would rephrase to “maternal and paternal, or more generally, parental effects). Rephrased (L 47)

L48: Replace “for” with “in”.

Done (L 48)

L49: Replace the first “and” with “within”.

Done (L 49)

L53: Add a comma after “effects” and after “circumstances”.

Done (L 53)

L61: Insert a bracket after “fruit flies”.

Done (L 61)

L65: I realise this was mentioned in the previous round of review – but “little stressful” doesn’t sound right. “Conditions with little stress” seems more appropriate.

Rephrased (L 64, 65)

L80: Change to “rather than simply parental information” or something similar.

Rephrased (L 80)

L86: Add a comma after “generations”.

Done (L 86)

L93-96: This sentence is rather long and a bit difficult to read. Rewrite.

Rephrased (L 93 to 96)

L115-116: This last line doesn’t flow well, I’d see if you could link it better to the previous sentence. Similarly on L126-127.

Both rephrased (L 115 to 117 and 127 to 128)

L160-162: I would remove the statistical text from this paragraph and create a new table with all of the relevant information and simply reference that, similar to above for consistency.

As suggested by the editor, the estimated marginal means are now provided in the supplementary material (supplementary table 6).

L168: Remove the third comma.

Rephrased (L 170)

L170: You might want to expand upon why it points to protein synthesis by specifically referring to the referenced papers in *Drosophila*, otherwise this point kind of appears out of nowhere.

Rephrased; the pertinent *Drosophila* reference has been present before (L 172)

→L345: “thrips naïve”?

Corrected (L 343)

Tables – I noticed in the response to reviewers that the notion of presenting effect sizes in tables was not considered. For ease of reading, and not having to work out the effect from the graph, it would better to

have the effect size and standard error in each table for each model. Particularly as the estimates from models will often not be similar to the raw data in light of random/fixed effects.

For each model, the estimated marginal means of significant main and interaction effects are now provided in the supplementary material (supplementary tables 1 to 6)

The full model outputs with appropriate model selection (which is mentioned in the methods but is not then available to review in the paper or in the SM – unless I've missed it) should then be placed in the supplementary.

Done (supplementary tables 1 to 6)

Figure 3: Replace Thrips_Rank with Thrips Rank.

Done